# When Does Group Invariant Learning Survive Spurious Correlations?

**Yimeng Chen**[1,2]*, **Ruibin Xiong**[3], **Zhiming Ma**[1,2], **Yanyan Lan**[4,5]†

[1]Academy of Mathematics and Systems Science, Chinese Academy of Sciences
[2]University of Chinese Academy of Sciences [3]Baidu Inc.
[4]Institute for AI Industry Research, Tsinghua University
[5]Beijing Academy of Artificial Intelligence, Beijing, China
`chenyimeng14@mails.ucas.ac.cn,xiongruibin@baidu.com,`
`mazm@amt.ac.cn,lanyanyan@tsinghua.edu.cn`

## Abstract

By inferring latent groups in the training data, recent works introduce invariant learning to the case where environment annotations are unavailable. Typically, learning group invariance under a majority/minority split is empirically shown to be effective in improving out-of-distribution generalization on many datasets. However, theoretical guarantee for these methods on learning invariant mechanisms is lacking. In this paper, we reveal the insufficiency of existing group invariant learning methods in preventing classifiers from depending on spurious correlations in the training set. Specifically, we propose two criteria on judging such sufficiency. Theoretically and empirically, we show that existing methods can violate both criteria and thus fail in generalizing to spurious correlation shifts. Motivated by this, we design a new group invariant learning method, which constructs groups with statistical independence tests, and reweights samples by group label proportion to meet the criteria. Experiments on both synthetic and real data demonstrate that the new method significantly outperforms existing group invariant learning methods in generalizing to spurious correlation shifts[1].

## 1   Introduction

In many real-world applications, machine learning models inevitably encounter data that are rarely presented in the training environment, i.e. being *out-of-distribution* (OOD). For example, data collected under new weather [36], locations [6], or light conditions [9] in vision tasks. However, machine learning models often fail in generalizing to OOD data, which blocks their deployment to critical applications [12; 38]. The dependence on spurious correlations that are prone to change across environments has been recognized as a major cause of such failure [4; 12; 37]. For example, it has been shown that models trained on MNLI [39] usually classify sentence pairs with high word overlap as the label 'entailment', regardless of their semantics [25]. On a new dataset where such relation no longer holds, the performance drops over 25% [25; 7].

A notable line of research on improving the robustness of models to distribution shifts is learning features with invariant conditioned label distribution across training environments [28; 4; 15], which has been termed *invariant learning* (IL). These methods are based on the assumption that the causal mechanism keeps invariant across environments [28], while the spurious correlation varies. By

---

*Work done during Yimeng Chen's internship at AIR, Tsinghua University.

†Corresponding author.

[1]Code is availiable at `https://github.com/Beastlyprime/group-invariant-learning`.

penalizing the variance of model prediction across environments, models are then encouraged to capture the causal mechanism instead of spurious correlations.

Recently, invariant learning has been introduced to the scenario where environment labels are unknown [33; 8], which we term the *group invariant learning* (group-IL). These methods utilize prior knowledge of spurious correlations to split the training data into groups. For example, Teney et al. [33] cluster training samples with their predefined spurious features. A more generic method, EIIL [8], splits training data into the majority/minority sets on which the spurious feature conditioned label distribution varies maximally. Similar to a priori environments, these groups are supposed to encode variations of spurious information, while holding the causal mechanism.

Though some performance improvements have been gained on several datasets, much uncertainty still exists on the effectiveness of group-IL methods. In particular, *when these methods can effectively address spurious correlations* remain a question. Though some theoretical analysis on the success and failure cases of IL with known environments has been proposed [2; 31; 21; 3], they are not sufficient for group-IL. First, inferred groups may not meet the assumptions on environments in existing theoretical analysis, thus their conclusions cannot generalize to group-IL. For example, in each environment, the spurious feature is assumed to have a Gaussian type distribution in [31]. However, the inferred group may not satisfy that condition, for example each group may only contain one unique value of spurious feature. Second, as environments are known and defined with causal structures, exiting analysis on the effect of environment on IL focus mostly on their number [31; 3] but less on their property or validity. However, the later are important for group-IL. Therefore, we need specific theoretical analysis under the setting of group-IL.

In this study, we discuss necessary conditions for group-IL to survive spurious correlations. For this purpose, we first formalize the setting of group-IL and clarify the necessary assumptions required for group-IL, which underlie our theoretical analysis. We then propose two criteria for group-IL, namely *falsity exposure criterion* and *label balance criterion*. They are respectively for judging whether spurious correlations are sufficiently exposed through their variation across groups and whether group invariance can reach spurious-free conditions. Based on that, we discuss the success and failure cases of existing methods. In some synthetic benchmarks (e.g. colored-MNIST in [4; 8]), the majority/minority groups meet the two criteria. However, in a case when the spurious feature is multivariate, the majority/minority split violates both criteria according to our theoretical analysis and observations on real datasets. As a result, existing group-IL methods are insufficient for solving spurious correlations.

To fix these problems, we propose a new group-IL method guided by the two criteria. Specifically, this method contains the following two steps to meet the two criteria. First, groups are defined by stratifying the prediction of a reference predictor which encodes spurious correlations. The strata is constructed with statistical tests, such that the spurious prediction is independent of the label on each group. Second, the label proportion of each inferred group is balanced by attaching weights to each instance within the group. Models are then trained with invariant learning objectives on the defined groups. We term this method Spurious-Correlation-Strata Invariant Learning with Label-balancing, abbreviated as SCILL. We further show that SCILL is provably sufficient in reaching spurious-free with ideal reference models.

To demonstrate the effectiveness of our proposed strategy, we conduct experiments on both synthetic and real data benchmarks on spurious correlations shifts in image classification and natural language inference (NLI). Specifically, we adopt two different invariant learning objectives, IRM (IRMv1) [4] and REx (V-REx) [15], to show the consistency of SCILL. To show the availability of SCILL, we also experiment with PGI [1] and cMMD [16; 1], which are feature invariance targets used with EIIL [8] in [1]. The experimental results show that SCILL with all the four invariance objectives consistently outperforms the existing state-of-the-art method EIIL in generalizing to spurious correlation shifts. Ablation study further shows the effectiveness of each component in SCILL.

Our main contributions can be summarized as follows.

- We propose two criteria for group-IL and analyze the insufficiency of existing methods.
- Guided by the two criteria, we propose a new practical group-IL method which is provably sufficient in solving spurious correlations.
- Extensive experiments on both synthetic and real-data benchmarks show that SCILL significantly outperforms existing methods on image classification and NLI tasks.

## 2   Related works

**Combating spurious correlations.**   A typical kind of distribution-shift is caused by the shift of spurious correlations [38], which are correlations between meaningless features (e.g. hospital tokens on a lung scan) and labels (e.g. has pneumonia or not) in the training set. The existence of spurious correlations in popular benchmarks have been revealed by many works [13; 29; 25; 32]. For example, predictive models with only incomplete semantic inputs or syntactic statistics can achieve high accuracy in NLI benchmarks [13; 29]. Geirhos et al. [11] point out that deep neural networks are prone to take easy-to-fit spurious correlations, i.e. *shortcut* strategies, in solving problems. As a result, resolving model's dependence on spurious correlation is important for their robustness to distribution shifts.

**Invariant learning.**   Many works on domain generalization focus on capturing invariancies across training environments [12]. Recently a new kind of strategy which has made significant impacts is to learn features that permit an invariant predictor across environments [28; 4; 15; 37], termed invariant learning in this paper. Such strategy is grounded upon the theory of causality [27], where Structural Equation Models [28] or causal graphs [37] are used to describe assumptions on the data generation process. The feature-conditioned label distribution invariance is then induced by the invariance of causal mechanisms in different environments. Recent theoretical works on IL study their failure cases in the domain generalization task where environments are known a priori [2; 31; 21; 3]. In this study, we focus on the setting of group-IL.

**Group invariant learning.**   In recent works, invariant learning is extended to the scenario without a priori environment labels, but with knowledge on spurious correlations in the training data [33; 8]. Such knowledge is proven to be necessary in this setting [17]. They are utilized to split the training data into groups, which are supposed to encode variations of spurious information so that they can be avoided by learning the invariance. For example, Teney et al. [33] cluster samples according to their question types. Liu et al. [19] construct groups with varying spurious correlations, based on the spurious features uncovered with feature selection. A more generic method proposed in EIIL [8] assume the access to a reference predictor which encodes spurious correlations, and exploit the outputs of the reference predictor to split training data into two groups, namely the majority and the minority. This strategy is shown effective, sometimes even outperform an oracle method using true environment labels in proving the OOD generalization [8] and also systematic generalization [1]. These empirical observations show the potential of group-IL methods, while also reveal the difference between the inferred groups and prior environments. As will be discussed in Section 3, group-IL methods need different assumptions on the causal structure from those for IL. Thus existing theoretical conclusions for IL can not generalize to group-IL. Therefore, we need specific theoretical analysis under the setting of group-IL. In a recent work, Lin et al. [17] derived the sufficient and necessary assumptions for their proposed algorithm. However, in this paper we focus on group criteria for general group-IL methods.

## 3   Formalization of group invariant learning

In this section, we introduce the problem setting and formalization of the scheme of group-IL.

Consider the task of learning a classifier $f : \mathcal{X} \to \mathcal{Y}$, which maps a value $x \in \mathcal{X}$ of the input variable $X$ to a value $y \in \mathcal{Y}$ of the target variable $Y$. For example, map an image of horse on grass to the label 'horse'. We denote $x_{inv}$ as the essential features of an instance $x$ of the input variable $X$ which define its class label (e.g. the shape of the horse), while $x_{sp}$ as features of $x$ should not inform the label of $x$ (e.g. the grass background). $X_{inv}$, $X_{sp}$ denote the corresponding random variables. The target is then to learn a *spurious-free* predictor whose predictions only depend on the feature $x_{inv}$, thus is supposed to have invariant performance on any dataset.

We first introduce the setting of IL. IL methods suppose that the training data $\mathcal{D}$ are collected under multiple environments $\mathcal{E}$, i.e. $\mathcal{D} = \{D_e\}_{e \in \mathcal{E}}$. $D_e = \{x_i^e, y_i^e\}_{i=1}^{n^e}$ contains data *i.i.d.* sampled from the probability distribution $\mathbb{P}^e(\mathcal{X} \times \mathcal{Y})$. $\mathbb{P}^e(Y|X_{inv})$ is invariant among different $e$, while $\mathbb{P}^e(Y|X_{sp})$ varies. Such an idea is based on the invariance of causal mechanisms across environments [28; 4], with assumptions on the causal structure of the data generating process. Figure 1 demonstrates four kinds of different assumptions in existing works on invariant learning. In these causal structures, environment

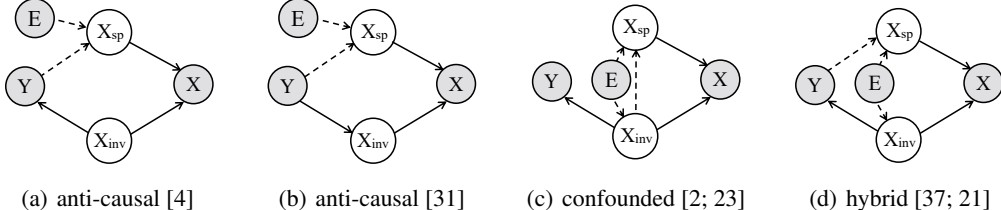

(a) anti-causal [4]    (b) anti-causal [31]    (c) confounded [2; 23]    (d) hybrid [37; 21]

Figure 1: The causal graph depicting different assumptions on the data generating process in existing IL works (some are simplified). Shading indicates the variable is observed. Dotted arrow indicates possible causal relation. The spurious feature is anti-causal in (a) and (b), confounded with the invariant feature in (c), and both anti-causal and confounded in (d).

is treated as a random variable $E$ take values in $\mathcal{E}_{all}$, which satisfies $\mathcal{E} \subset \mathcal{E}_{all}$. $\mathbb{P}^e(\cdot) := \mathbb{P}(\cdot|E = e)$. $X_{inv}$ and $X_{sp}$ are latent feature variables generating the observation $X$, i.e. $X = r(X_{inv}, X_{sp})$. Here $r$ is generally assumed as a bijective function so that the latent features can be recovered from the observations [31; 2; 37]. Align with the formalization in the former paragraph where $X_{sp}, X_{inv}$ are assumed recognizable from $X$, in this paper we adopt the same assumption. In all the four kinds of causal graphs, $\mathbb{P}^e(Y|X_{inv}) := \mathbb{P}(Y|X_{inv}, E = e)$ keeps invariant under different $e \in \mathcal{E}_{all}$, while $\mathbb{P}^e(Y|X_{sp})$, $\mathbb{P}^e(X_{sp})$, $\mathbb{P}^e(X_{inv})$, and $\mathbb{P}^e(X_{inv}|Y)$ can vary across different $e \in \mathcal{E}_{all}$.

Suppose the predictor $f$ can be decomposed into $f = c \circ \Phi$, where $\Phi : \mathcal{X} \to \mathcal{H}$ denotes a feature encoder which maps the input into a representation space $\mathcal{H}$, $c : \mathcal{H} \to \mathcal{Y}$ is a classifier. The target of invariant learning is then to search for a $\Phi$ which satisfies the following constraint:

$$\mathbb{P}(Y|\Phi(X), E = e) = \mathbb{P}(Y|\Phi(X), E = e'), \forall e, e' \in \mathcal{E}. \tag{EIC}$$

It is termed as *Environment Invariance Constraint* (EIC). Note that the EIC stated in [8] is a weaker form of this EIC. The constraint is incorporated into the training target via a penalty term. In a generic form, the learning target of invariant learning methods can be written as follows:

$$\min_f \sum_{e \in \mathcal{E}} \lambda_e \mathcal{R}^e(f) + \lambda \cdot penalty(\{S_e(f)\}_{e \in \mathcal{E}}) \tag{1}$$

where $\mathcal{R}^e(f)$ stands for the expected loss of $f$ on the environment $e$, weighted by a scalar $\lambda_e$. $S_e(f)$ stands for some statistics of $f$ on $e$, and the *penalty* is on the variation of $S_e(f)$ to measure the deviation degree of EIC. In IRM [4], the penalty is the summation of $S_e(f) = \|\nabla_w \mathcal{R}^e(w \circ f)\|^2$, where $w$ is a constant scalar multiplier of 1.0 for each output dimension. In V-REx [15], $S_e(f) = \mathcal{R}^e(f)$, and the penalty is the variance of $S_e(f)$ on different environments. In CLOvE [37], the penalty is defined as the summation of calibration errors of the model on each environment.

Group invariant learning methods release the dependency of invariant learning on predefined environments by splitting the training data into groups. Suppose $\mathcal{D}$ is sampled from the distribution $\mathbb{P}(\mathcal{X} \times \mathcal{Y})$. Intuitively, those groups are expected to hold the invariant mechanism $\mathbb{P}(Y|X_{inv})$, while informing the variation of $X_{sp}$. As a result, it is meaningless to divide groups according to $X_{inv}$. In group-IL methods [33; 19; 8], group inference algorithms are designed to utilize knowledge on $X_{sp}$ or the correlation between $X_{sp}$ and $Y$. Formally, denote the inferred groups as $\mathcal{G} := \{g_1, g_2, \ldots, g_m\}$. Define $G : \mathcal{X} \times \mathcal{Y} \to \mathcal{I}$ as the function which maps a sample to its group identity, i.e. $G(x, y) = i$ if and only if $(x, y) \in g_i$. $\mathcal{G}$ is then the set of events $\{G = i\}, i \in \mathcal{I}$. We have $G$ is $\sigma(X_{sp}, Y)$-measurable, i.e. it is a function of $X_{sp}$, as in [33; 19], or both $X_{sp}$ and $Y$ [8].

In the following sections, we ground our analysis on group-IL with the causal structures in Figure 1 (a) and (b), while without additional assumptions on the causal models. Our choice of causal structures is based on the following two observations. First, in the causal graph (d), $X_{sp}$ is a backdoor variable [27] between $X_{inv}$ and $Y$ and confounded with $X_{inv}$ by an unobserved variable. As a result, whether the invariant mechanism holds on each group is indeterminate without additional assumptions on the mechanisms between $X_{sp}$ and $X_{inv}$. Second, as shown by Ahuja et al. [2], invariance itself can not deal with spurious feature for causal structure (c). Additional knowledge or penalty, e.g. information bottleneck, is needed together with group-IL. Therefore, we investigate group-IL under the causal structures depicted by graphs (a) and (b).

# 4 Two group criteria

With the above formulation, we are ready to theoretically analyze the ability of existing group-IL methods in learning spurious-free predictors. For this purpose, in this section we derive two necessary conditions for group-IL in surviving spurious correlations, i.e. falsity exposure, and label balance. Both conditions can be used as criteria to judge the sufficiency of group-IL methods. We then show that existing methods can violate the two criteria, thus become insufficient for learning a spurious-free predictor.

## 4.1 Falsity exposure criterion

As groups are supposed to expose variance of spurious features so that they can be avoided by invariant learning, a natural idea is to take into account the sufficiency of such exposure on inferred groups. Ideally, if groups are split according to $X_{sp}$, any variance of $\mathbb{P}(Y|X_{sp})$ is then fully exposed. However, such split is only practical when the value of $X_{sp}$ is accessible and sparse. On the contrary, we consider the condition when groups provide insufficient exposure. Intuitively, if some spurious correlation keeps invariant across groups, its variation is then not exposed, thus group invariant predictor may still depend on such correlation. Formally, this can be written as the following criterion.

**Criterion 4.1** (Falsity Exposure). For any $\sigma(X_{sp})$-measurable function $h$ that satisfies $\forall g, g' \in \mathcal{G}$, $\mathbb{P}(Y|h(X_{sp}), g) = \mathbb{P}(Y|h(X_{sp}), g')$, it must satisfies $\mathbb{P}(Y|h(X_{sp})) = \mathbb{P}(Y)$.

Intuitively, if the falsity exposure criterion is not satisfied, $h(X_{sp})$ will be an invariant feature across environments, with predictive ability on $Y$. The predictor depending on both $X_{inv}$ and $h(X_{sp})$ can satisfy EIC but fails to be free of spurious features. The following theorem formalizes the significance of the falsity exposure criterion.

**Theorem 4.2.** *Suppose the falsity exposure criterion is violated, i.e. $\exists h$ which satisfies $\mathbb{P}(Y|h(X_{sp}), g) = \mathbb{P}(Y|h(X_{sp}), g') \neq \mathbb{P}(Y), \forall g, g' \in \mathcal{G}$. Then the optimal solution of group-IL is $f(X) = \mathbb{P}[Y|X_{inv}, h(X_{sp})]$, which fails to generalize when $\mathbb{P}(Y|X_{sp})$ shifts.*

## 4.2 Label balance criterion

Even if we have sufficient falsity exposure, would the constraint in group-IL, i.e. EIC, guarantee the model to be free of spurious correlations? We study this problem by analyzing the relation between EIC and the constraint for a predictor to be spurious-free. In our assumed causal structures, $X_{inv} \perp\!\!\!\perp X_{sp}|Y$. Thus, a spurious-free predictor, which only depends on $X_{inv}$, satisfies $f(X) \perp\!\!\!\perp X_{sp}|Y$. In fact, it can be proved that this is a sufficient condition for a predictor $f(X)$ to be invariant to the intervention [27] on $X_{sp}$ (See the appendix). As a result, we term $f(X) \perp\!\!\!\perp X_{sp}|Y$ as the *spurious-free constraint*, i.e.

$$\mathbb{P}(f(X)|X_{sp} = b, Y) = \mathbb{P}(f(X)|X_{sp} = b', Y), \forall b, b' \in \mathcal{B}, \tag{SFC}$$

where $\mathcal{B}$ is the image set of $X_{sp}$. The following is a necessary condition for EIC to induce the above constraint, which we term as the *label balance* criterion. It states that the label proportion in different groups should be the same.

**Criterion 4.3** (Label Balance). For any $g, g' \in \mathcal{G}$ and $y, y' \in \mathcal{Y}$ with non-zero $\mathbb{P}(Y = y|g), \mathbb{P}(Y = y'|g), \mathbb{P}(Y = y|g')$ and $\mathbb{P}(Y = y'|g')$, the following equation holds.

$$\mathbb{P}(Y = y|g)/\mathbb{P}(Y = y'|g) = \mathbb{P}(Y = y|g')/\mathbb{P}(Y = y'|g') \tag{2}$$

Formally, the following theorem shows the significance of this criterion on the effectiveness of group-IL.

**Theorem 4.4.** *With a set of groups $\mathcal{G}$ inferred by $(X_{sp}, Y)$, i.e. $\mathcal{G} \subset \sigma(X_{sp}, Y)$, if the label balance criterion is violated, functions satisfying EIC can not satisfy SFC.*

## 4.3 Analysis of existing group invariant learning methods

Now we analyze whether the groups in existing group-IL methods meet the two criteria. Directly, randomly grouped clusters of $X_{sp}$ as in [33] do not guarantee to meet either criteria, and clusters of

$\mathbb{P}(Y|X_{sp})$ [19] do not meet the label balance criterion. In the following discussions, we focus on the majority/minority groups inferred by the EI algorithm in EIIL [8]. We find that in some synthetic datasets where the spurious feature only has two distinct values, the majority/minority groups satisfy the two criteria. However, observation on real datasets and theoretical results on the case when the spurious feature is multivariate show that they can violate both criteria.

On some synthetic datasets constructed in existing works [1; 8], the majority/minority groups satisfy these two criteria. For example, on both colored-MNIST [4] and coloured-MNIST [1], $Y$ has a uniform distribution, and the spurious correlation has the same ratio for any spurious features, e.g. $\mathbb{P}(Y = 0|\text{color} = green) = \mathbb{P}(Y = 1|\text{color} = red)$ on colored-MNIST. It can be proved that in this case, the majority/minority groups satisfy both criteria (See the appendix).

However, it no longer holds in general cases. Empirically, we observe that on MNLI, the label distributions of the two groups inferred by EI are significantly different. Specifically, the ratio of the counts of label 0 and label 1 in the two groups are 2.87 and 0.17 respectively. The following proposition theoretically provides a case where the majority/minority split satisfies the label balance but breaks the falsity exposure, and invariant learning objectives fail to find the spurious-free classifier.

**Proposition 4.5.** *Suppose we have $(X, Y) \sim \mathbb{P}(X, Y)$. $Y$ takes value in $\{0, 1\}$. $X$ is formed with spurious feature variable $X_{sp} = (B_0, B_1)$, and invariant feature variable $S$, i.e. $X = r(B_0, B_1, S)$, for some bijective function $r$. $B_0$ and $B_1$ are both binary variables, which take values in $\{b_0^0, b_0^1\}$ and $\{b_1^0, b_1^1\}$ respectively. $B_0$, $B_1$ and $S$ are conditionally independent given $Y$. Suppose $\mathbb{P}(Y = j|B_i = b_i^j) = p_i, \forall i, j \in \{0, 1\}$, and $p_0 > p_1$. Then we have 1) the majority/minority groups $e_{mar}, e_{min}$ violate the falsity exposure criterion. 2) the optimal classifier under invariant learning objectives depends on $B_1$.*

In this case, we suppose the spurious feature can be decomposed into two variables that are conditionally independent with each other given the label. Such case can realize when the dataset contains multiple kinds of spurious features. For example, in the image classification task, the background pattern and the color of an object can be independent but both correlate with the label spuriously.

# 5 SCILL: a new method

According to the above discussion in Section 4, existing group-IL methods may fail to meet the two criteria, leading to insufficient training loss for solving spurious correlations. Faced with this challenge, we propose a new group-IL method, to satisfy the two criteria. For the generality, we only assume the access to a reference model, as in EIIL, instead of spurious features. Our new method includes two steps. In the group inference step, we define groups as spurious correlation strata constructed with the reference model, for the falsity exposure criterion. While in the training objective step, we reweight each sample with group label proportion to meet the label balance criterion. To highlight the two parts, our method is named as Spurious-Correlation-strata Invariant Learning with Label-balance, abbreviated as SCILL. Theoretically, we prove that this method is provably sufficient with ideal reference models, i.e. its optimal solution is spurious-free.

## 5.1 Group inference with statistical split

We first introduce the group inference step in SCILL. As in EIIL, we assume a reference classifier $f_r$ is available, which is expected to predict only depending on $X_{sp}$. $f_r$ can be a model trained with empirical risk minimization (ERM) [8] or a model designed to capture spurious correlations [20].

We propose to construct groups $\mathcal{G}$ by stratify the outputs of $f_r$, with the target that $Y \perp\!\!\!\perp f_r(X)|g$, $\forall g \in \mathcal{G}$. The motivation for introducing spurious correlation strata comes from the deduction of the falsity exposure criterion, i.e. when groups meet the requirements that $X_{sp} \perp\!\!\!\perp Y|g$, the falsity exposure criterion is satisfied. That is because with the above condition, we have for any function $h$, $h(X_{sp}) \perp\!\!\!\perp Y|g$, as a result $\mathbb{P}(Y|h(X_{sp}), g) = \mathbb{P}(Y|g)$. If $\mathbb{P}(Y|h(X_{sp}), g) = \mathbb{P}(Y|h(X_{sp}), g')$, $\forall g, g' \in \mathcal{G}$, we have $\mathbb{P}(Y|g) = \mathbb{P}(Y|g') = \mathbb{P}(Y)$. Thus $\mathbb{P}(Y|h(X_{sp}), g) = \mathbb{P}(Y)$. As a result, the falsity exposure criterion is satisfied. As different output values of the reference model $f_r$ inform the difference in $\mathbb{P}(Y|X_{sp})$, $Y \perp\!\!\!\perp f_r(X)|g$ approximates $\mathbb{P}(Y|X_{sp}) \perp\!\!\!\perp Y|g$, which is equivalent to $X_{sp} \perp\!\!\!\perp Y|g$.

We propose to construct such groups through the *statistical-split* algorithm, inspired by the algorithm proposed in [14] for propensity score estimation. Specifically, we split current groups into subgroups according to the hypothesis-test statistics. For example, for the binary classification case, a sample set $B$ is first divided into two subsets $L_0$, $L_1$ according to the labels of samples. Then the two-sample t-statistics $t_B$ of $\log[f_r(x)_0/f_r(x)_1]$ are computed on the two sets. If $t_B$ exceeds a fixed threshold $thr$, $B$ is then split into two subsets according to the median of $f_r(x)_0$ on $B$. In this way, we enhance $f_r(X) \perp\!\!\!\perp Y$ in each group. Note that $thr$ is a hyperparameter of this algorithm. Empirical robustness analysis on $thr$ is conducted in our experiments. More details about the algorithm and the robustness study are provided in the appendix.

## 5.2 Training with reweighted loss

Now we introduce the second step. To guarantee the label balance criterion, we reweight each sample with group label proportion in the training loss. Correspondingly, the objective of invariant learning becomes the following form:

$$\mathcal{L}(f) := \sum_{g \in \mathcal{G}} \tilde{\mathcal{R}}^g(f) + \lambda \cdot penalty(\{S_g(f)\}_{g \in \mathcal{G}}) \tag{3}$$

where $\tilde{\mathcal{R}}^g(f) = \mathbb{E}[w^g(Y)\mathcal{L}^g(f(X), Y)], \omega^g(y) := \mathbb{P}(Y = y)/\mathbb{P}(Y = y|g)$ for nonzero $\mathbb{P}(Y = y|g)$. The weight function is defined to balance the label distribution between groups, i.e. $\mathbb{P}(Y|g) = \mathbb{P}(Y|g'), \forall g, g'$, for achieving the label balance criterion.

With the above two steps, SCILL is then able to meet the two group criteria. Now we further investigate the theoretical capability of SCILL in solving spurious correlations. The following theorem shows that with a purely spurious reference model, SCILL can find spurious-free predictors.

**Theorem 5.1.** *If $\mathcal{G}$ satisfies $f_r^*(X) \perp\!\!\!\perp Y|g, \forall g \in \mathcal{G}$, where $f_r^* : \mathcal{X} \to \mathcal{Y}$ is spurious-only, i.e. $\sigma(X_{sp})$-measurable, and minimizes the prediction loss $\mathcal{L}_{ce}^r = \mathbb{E}[\sum_y \mathbb{P}(Y = y|X) \log f_r(X)_y]$, the optimal model minimizing the objective (3) satisfies SFC.*

# 6 Experiments

In this section, we first conduct experiments to show that SCILL outperforms existing group-IL methods in generalizing to spurious correlation shifts. Then we empirically analyze whether the experimental improvements are consistent with the theoretical findings.

## 6.1 Experimental settings

Now we describe our experimental settings, including datasets, models, and some training details. More details are provided in the appendix.

**Datasets** We conduct experiments on both synthetic and real-world datasets. The synthetic dataset, Patched-Colored MNIST (PC-MNIST), is constructed as a realization of the conditions in the Proposition 4.5 to verify the proposed criteria. It is derived from MNIST, by assigning two conditionally independent spurious features given label, namely the color and patch bias to each image. The design of the patch bias is inspired by [5]. MNLI-HANS is a benchmark widely used in many previous works on combating spurious correlations, such as [7; 34]. In our experiments, we follow the practice to utilize MNLI [39] as the training data and HANS [25] as the test data.

**Baselines and configurations** In our experiments, we compare SCILL with two baselines, i.e. ERM and EIIL [8]. ERM represents the method with the traditional empirical risk minimization (ERM) approach. EIIL is a state-of-the-art group-IL method, where groups are inferred by searching an assignment to make the reference model maximally violates the invariant learning principle. We experiment with four different invariance penalties: IRM [4], REx (V-REx) [15], cMMD [16; 1] and PGI [1]. Note that cMMD and PGI target to learn group invariant predictions conditioning on the label, different from EIC. See Appendix for more details of the four penalties.

The training configurations are presented as follows. For PC-MNIST, we adopt the classifier proposed in [4] for Colored MNIST, which is a MLP with two hidden layers of 390 neurons. The reference

Table 1: Classification accuracy on PC-MNIST under three model selection strategies ID, Oracle, TEV. Val columns contain the accuracy values computed on the in-distribution validation set, and Test columns contain those on the test set. As the label noise rate is set to $0.25$ on PC-MNIST, the optimum predictor depending on invariant features achieves an accuracy around 75% on both sets.

| Method | Penalty | ID | | Oracle | | TEV | |
|---|---|---|---|---|---|---|---|
| | | Val | Test | Val | Test | Val | Test |
| ERM | - | 90.22 ± 0.56 | 50.64 ± 0.56 | 89.95 ± 0.45 | 54.53 ± 0.60 | - | - |
| EIIL | IRM | 90.21 ± 0.48 | 50.63 ± 0.45 | 78.01 ± 0.45 | 63.63 ± 0.71 | 69.81 ± 0.27 | 50.99 ± 0.58 |
| | REx | 90.24 ± 0.45 | 51.21 ± 0.64 | 79.10 ± 0.43 | 64.04 ± 0.80 | 70.05 ± 0.23 | 51.01 ± 0.68 |
| | cMMD | 90.24 ± 0.43 | 51.36 ± 0.61 | 77.27 ± 0.28 | 65.09 ± 0.63 | 70.15 ± 0.25 | 52.70 ± 1.40 |
| | PGI | 90.19 ± 0.46 | 51.07 ± 0.54 | 80.03 ± 1.41 | 64.27 ± 0.26 | 70.37 ± 0.14 | 50.64 ± 0.38 |
| SCILL | IRM | 79.65 ± 0.76 | 62.49 ± 0.55 | 71.54 ± 0.35 | 67.46 ± 0.19 | 71.54 ± 0.35 | 67.46 ± 0.19 |
| | REx | 80.23 ± 0.83 | 62.13 ± 0.99 | 72.59 ± 1.44 | **67.60** ± 0.24 | 70.77 ± 0.50 | 67.33 ± 0.30 |
| | cMMD | 83.13 ± 0.93 | 59.76 ± 0.92 | 73.12 ± 0.47 | 67.49 ± 0.52 | 72.38 ± 0.51 | **67.81** ± 0.34 |
| | PGI | 80.67 ± 1.75 | **62.52** ± 0.32 | 71.73 ± 1.43 | 67.26 ± 0.14 | 71.35 ± 0.24 | 67.36 ± 0.33 |
| SCILLuw | IRM | 90.27 ± 0.39 | 50.95 ± 0.47 | 90.07 ± 0.34 | 53.51 ± 1.38 | 90.28 ± 0.39 | 50.85 ± 0.47 |
| maj./min. | | 90.18 ± 0.26 | 50.67 ± 0.15 | 80.10 ± 0.21 | 63.85 ± 0.58 | 90.18 ± 0.26 | 50.67 ± 0.15 |
| SCILLgt | IRM | 82.55 ± 0.28 | 61.12 ± 1.17 | 74.46 ± 0.25 | 70.19 ± 0.39 | 72.30 ± 0.40 | 70.91 ± 0.06 |
| SCILLub | | 84.37 ± 0.53 | 58.78 ± 0.41 | 79.27 ± 2.95 | 59.44 ± 0.44 | 66.07 ± 0.73 | 56.20 ± 0.57 |
| opt | - | 75 | 75 | 75 | 75 | 75 | 75 |

Table 2: Classification accuracy on HANS.

| Method | Penalty | ID | | Oracle | | TEV | |
|---|---|---|---|---|---|---|---|
| | | Val | Test | Val | Test | Val | Test |
| ERM | - | 84.12 ± 0.15 | 64.88 ± 3.00 | 84.12 ± 0.15 | 64.88 ± 3.00 | - | - |
| EIIL | IRM | 84.01 ± 0.08 | 65.35 ± 0.93 | 83.82 ± 0.17 | 66.42 ± 0.98 | 84.01 ± 0.08 | 65.35 ± 0.93 |
| | REx | 84.10 ± 0.13 | 65.16 ± 0.19 | 83.91 ± 0.20 | 66.87 ± 2.92 | 84.00 ± 0.48 | 66.43 ± 1.00 |
| | cMMD | 83.56 ± 0.03 | 63.22 ± 1.76 | 83.22 ± 0.13 | 64.25 ± 1.63 | 83.38 ± 0.20 | 62.72 ± 2.03 |
| | PGI | 84.17 ± 0.08 | 65.57 ± 2.25 | 83.78 ± 0.03 | 66.02 ± 0.93 | 83.94 ± 0.64 | 65.57 ± 2.25 |
| SCILL | IRM | 82.75 ± 0.17 | 69.11 ± 1.76 | 82.56 ± 0.33 | 68.72 ± 1.24 | 82.67 ± 0.14 | 69.82 ± 1.29 |
| | REx | 82.68 ± 0.28 | **69.73** ± 1.63 | 82.59 ± 0.22 | **71.20** ± 1.81 | 82.56 ± 0.33 | 69.75 ± 1.53 |
| | cMMD | 82.74 ± 0.26 | 69.15 ± 1.39 | 82.39 ± 0.45 | 70.77 ± 1.40 | 82.61 ± 0.04 | **70.92** ± 0.79 |
| | PGI | 82.79 ± 0.30 | 68.57 ± 0.54 | 81.69 ± 0.28 | 70.99 ± 0.48 | 82.79 ± 0.30 | 68.57 ± 0.54 |
| EIILlb | IRM | 83.39 ± 0.06 | 63.90 ± 1.16 | 83.39 ± 0.06 | 63.90 ± 1.16 | 83.16 ± 0.22 | 61.33 ± 0.33 |
| SCILLuw | IRM | 84.15 ± 0.11 | 64.30 ± 0.67 | 83.77 ± 0.15 | 65.93 ± 0.12 | 83.84 ± 0.02 | 65.63 ± 1.46 |

model is a MLP with the same structure trained with ERM on the training set, following the setting in EIIL on Colored MNIST. While for MNLI, we use a BERT-based classifier with the standard setup for sentence pair classification [10]. The reference model is the same as the biased classifier propose in [34], which is trained on top of some hand-crafted syntactic features. For each task, all implementations of SCILL and EIIL adopt the same model configurations and pretrained reference models. Since models are tested with OOD data, it is important to specify the model selection strategy, as has been revealed by Gulrajani and Lopez-Paz [12] for the case of domain generalization. In our experiments, we report results with 3 different model selection strategies, including ID, Oracle, and TEV. ID refers to the strategy based on model performance on the in-distribution validation set as used in [34]. Oracle refers to the selection based on data from the test data distribution, as used in [8; 12]. While TEV is a new strategy adapted from the training-domain validation method in [12] to the inferred groups, which alleviates the dependence on the test data as ID. Details can be found in the appendix.

Table 3: Robustness study on PC-MNIST. It shows the performance of SCILL-IRM with threshold $5, 10, 15, 20$ on t-statistics in the statistical split algorithm. Top 2 values are in bold. Results in Table 1 are all under threshold 10.

| Method | #G | ID | | Oracle | | TEV | |
|--------|-----|-----|------|--------|------|-----|------|
| | | Val | Test | Val | Test | Val | Test |
| ERM | - | $90.22 \pm 0.56$ | $50.64 \pm 0.56$ | $89.95 \pm 0.45$ | $54.53 \pm 0.60$ | - | - |
| SCILL-thr-20 | 6 | $83.15 \pm 0.47$ | $\mathbf{60.14} \pm 1.12$ | $73.37 \pm 0.65$ | $\mathbf{67.95} \pm 0.66$ | $72.59 \pm 0.33$ | $\mathbf{67.79} \pm 0.57$ |
| SCILL-thr-15 | 7 | $82.84 \pm 0.61$ | $59.79 \pm 1.00$ | $73.07 \pm 0.69$ | $\mathbf{68.17} \pm 0.56$ | $72.31 \pm 0.32$ | $\mathbf{67.87} \pm 0.37$ |
| SCILL-thr-10 | 9 | $79.65 \pm 0.76$ | $\mathbf{62.49} \pm 0.55$ | $71.54 \pm 0.35$ | $67.46 \pm 0.19$ | $71.54 \pm 0.35$ | $67.46 \pm 0.19$ |
| SCILL-thr-5 | 15 | $76.91 \pm 0.60$ | $55.50 \pm 1.78$ | $66.29 \pm 13.1$ | $58.81 \pm 2.35$ | $60.29 \pm 9.97$ | $61.89 \pm 3.96$ |

## 6.2 Experimental results

Now we demonstrate our experimental results, including performance comparison and detailed analysis. More empirical results including the robustness analysis on the hyperparameter in SCILL can be found in the appendix.

### 6.2.1 Performance comparison

Table 1 and 2 show the experimental results on PC-MNIST and MNLI-HANS, respectively. The main observation is that all implementations of SCILL consistently outperform the counterpart of EIIL across all model selection strategies, in terms of the performance on OOD data. Comparing different model selection strategies, Oracle performs the best for both EIIL and SCILL. However, SCILL with the TEV strategy has the ability to outperform EIIL with Oracle, demonstrating the superiority of our new objective. Additionally, SCILL also gains improvements against some debiasing methods utilizing the same reference model (See the appendix).

### 6.2.2 Ablation study and verification of the two criteria

Our main theoretical results in Section 4, i.e. Theorem 4.2 and 4.4, reveal that the two group criteria are necessary conditions for group-IL to survive spurious correlations. Now we discuss the empirical verification of the significance of the two criteria.

**Falsity exposure criterion.** To show the significance of the falsity exposure criterion, we compare the performance of methods under the case when the label balance criterion is satisfied. On PC-MNIST, both SCILL and EIIL groups satisfy the label balance criterion[3], while EIIL groups provably violate the falsity exposure, according to Proposition 4.5. The significant improvement of SCILL over EIIL on Table 1 then shows the importance of the falsity exposure criterion. To exclude the effect of the noise in the reference model in the group inference, we further implement SCILL with the ground-truth spurious predictor, obtaining $SCILL_{gt}$ in Table 1. The groups then satisfy the falsity exposure criterion. We construct the ground truth majority/minority split and experiment with IL methods, obtaining results in the row maj./min. in Table 1. The significant performance drop of maj./min. compared with $SCILL_{gt}$ verifies the importance of falsity exposure for group-IL. On MNLI, the label in EIIL groups is unbalanced (See the appendix). Therefore we attach the instance reweight step as in SCILL to EIIL, obtaining $EIIL_{lb}$ which satisfies the label balance criterion. As shown in Table 2, $EIIL_{lb}$ fails to achieve improved performance, which verifies the necessity of falsity exposure.

**Label balance criterion.** To verify the necessity of the label balance criterion, we investigate the cases when the falsity exposure is satisfied. As the $SCILL_{gt}$ on PC-MNIST satisfies the falsity exposure, we construct such cases by disturbing the label balancing weights in SCILL. We multiply the estimated label proportion of class 0 by 0.5 to get the unbalanced $SCILL_{ub}$. As shown in Table 1, under Oracle selection, the test accuracy of $SCILL_{ub}$ drops approximately 15% compared with $SCILL_{gt}$, thus verifying the impact of label balance. More results can be found in the appendix.

---

[3]The ratio of label 0 and label 1 on group 0 and group 1 in EIIL is 1:1.03 and 1:1.04, respectively.

We further show the importance of the instance reweight step in SCILL, which is designed following the label balance criterion. For this, we remove the instance reweight step in SCILL, obtaining SCILLuw. The experimental results in Table 1 and 2 show that SCILLuw performs worse than SCILL, demonstrating the importance of the instance reweight step in SCILL.

### 6.2.3 Robustness analysis

As shown in Section 5.1, the statistical-split algorithm contains a hyper-parameter $thr$. So We study the robustness of SCILL w.r.t. $thr$ by experiments on PC-MNIST with $thr$ set as $5, 10, 15, 20$ in SCILL-IRM. From the results shown in Table 3, the models are robust with different $thr = 10, 15, 20$, though the model with $thr = 5$ is worse than others. More results can be found in the Appendix.

## 7 Discussions

The main limitation of the paper is our assumptions on the causal structure. In fact, our conclusions can be generalized to more complex structures, e.g. those in [35] (See the appendix). The most central assumption is the conditional independence between $X_{inv}$ and $X_{sp}$ given $Y$, which is adopted in many existing works on solving spurious correlations [35; 7; 40; 30]. It would be an important direction to find causal structures on which the assumption is not satisfied while group invariant learning can still be effective.

In this paper, the algorithm SCIIL is proposed as a possible but not necessarily optimal solution to meet the two criteria for group-IL. As this paper focuses on analyzing group invariant learning, comparing SCILL with other algorithms besides group-IL is beyond the scope of this paper. It is noteworthy that the objective function of SCIIL exhibits similarities to those utilized in two recent methods. [24; 30]. Both methods incorporate a risk term reweighted by estimations of spurious correlations and a feature invariance penalty. However, these methods are limited to scenarios when spurious features can be explicitly defined [30], and are also discrete as assumed in [24]. Additionally, their feature invariance penalty differs from those employed in IL. Compared to some other methods that utilize a reference model [7; 22; 34; 26; 18; 40], the first term of the SCIIL algorithm is in similar form as it also involves reweighting samples based on the outputs of the reference model. However, the IL penalty in SCIIL serves as an additional regularization term. An extended discussion is provided in the appendix.

Besides the setting of group invariant learning, the two criteria may also bring benefits to the study of domain generalization. Furthermore, our empirical results show SCILL can achieve good performance with other kinds of invariances besides that in invariant learning, e.g. the invariance of the distribution of model outputs conditioned on the label. Discussing the effect of the two group criteria on other kinds of group invariance is a potential research direction.

## 8 Conclusion

This paper is concerned with when group invariant learning (group-IL) can survive spurious correlations. We first formulate the setting of group-IL and necessary assumptions. Then we theoretically analyze the necessary conditions for group-IL in learning spurious-free predictors, and obtain two group criteria, i.e. falsity exposure and label balance. Considering the limitations of previous group-IL methods, we propose a new method SCILL to satisfy the two criteria. Furthermore, we theoretically prove that SCILL has the ability to learn a spurious-free predictor. Finally, we conduct extensive experiments on both synthetic and real data to evaluate the proposed new method. Experimental results show that SCILL significantly outperforms existing SOTA group-IL methods, owing to its ability to satisfy the two criteria. The empirical studies validate our theoretical findings.

## 9 Acknowledgement

This work was supported by National Key R&D Program of China No. 2021YFF1201600, Vanke Special Fund for Public Health and Health Discipline Development, Tsinghua University (NO.20221080053), and Beijing Academy of Artificial Intelligence (BAAI). The authors would like to thank Keyue Qiu and Yuan Li for providing useful feedback on the draft.

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
