# OpenReview forum: "When Does Group Invariant Learning Survive Spurious Correlations?"
_NeurIPS.cc/2022/Conference — NeurIPS 2022 Accept_

### Official Review · Reviewer_i5VX · 2022-07-10

**Rating:** 5
**Confidence:** 4
**Soundness:** 2 fair
**Presentation:** 2 fair
**Contribution:** 3 good

**Summary:**

This paper proposes and evaluates conditions for group invariant learning methods in preventing classifiers from depending on spurious correlations. Motivated by the proposed conditions, the authors develop a new group learning approach with a two-step procedure involving constructing group structures to achieve these conditions and re-weighting samples in model training according to group information.

**Questions:**

line 123: please clarify definition of $D'$

line 186: please give the definition of $B$, even if appeared in the appendix

Please clarify in the main text the relation between $X$ and $X_{inv}$ and $X_{sp}$. e.g. whether $X$ is generated as a mapping from $(X_{inv}, X_{sp})$

How strong is the condition in Criterion 4.3 and how would the results be affected if the criterion is violated?




**Limitations:**

Yes

**Strengths And Weaknesses:**

This paper studies an important question regarding the effectiveness of group invariant learning at the presence of spurious features. The proposed method is compared in the experiment with a good variety of methods and datasets and has demonstrated potential to improve current methods.

However, there are some issues in the technical results and presentations that prevent the readers from fully following the results. For example, In Proposition 4.5, line 217, it is not quite clear why $P(Y=j | B_i) = p_i$, where the right side depends on $i$ alone. In line 215, $r$ is not defined although its meaning can be vaguely guessed. Similarly, in line 260, the $\mathbb{G}$ is not defined. Another common issue is the usage of conditional probability/distributions given group $g$, which seems to be really referring to conditional on the event $\{G=g\}$, where $G$ is the random mapping from the sample $(X, Y)$ to the deterministic group label $\mathcal{G}$ in which $g$ is an element.

What is more important than the notation issues is that the main results of this paper seem to center around the construction of the random mapping $G$ as a function of $Y$, $X_{inv}$, and $X_{sp}$ to achieve the independent criterions for group invariant learning to work, but the current presentation of results does not completely bridge the gap between the criterion and the final conclusion and nail the requirement of $G$ in terms of the spurious features. Since the probability model framework plays a central role in this paper, it would be helpful to make sure the arguments are rigorous and seamless.

The paper would also benefit from an ablation study to check the importance of the two proposed criteria as well as varying arguments that affect the results including the grouping $\mathcal{G}$ and the weights in the objective $w$.

---

> ### Author Response · Authors · 2022-08-02
> **Response to Reviewer i5VX**
>
> We appreciate the effort the reviewer has put into their feedback and agree that the points raised by the reviewer deserve further clarification. We are pleased to note the positive mark on the contribution of this work. We provide below a detailed response for each of the points that were raised and have revised the paper accordingly.
>
> **- The definition of some notations, including [line 123] D', [line 186] B, [line 215] r, [line 290] G, and clarify the relation between $X$ and $X_{inv}$ and $X_{sp}$.**
>
> We thank the reviewer for pointing these out. We have revised such notation issues including some typos in the new revision. Among them, $\mathbb{G}$ in Line 290 is a typo. It should be $\mathbb{E}$ which denotes the expectation.
>
> **- In Proposition 4.5, line 217, it is not quite clear why $P(Y=j|B_i)=p_i$, where the right side depends on $i$ alone.**
>
> The equation in Line 217 is a condition, which writes as $\mathbb{P}(Y=j|B_i = b_i^j) = p_i$, i.e. $\mathbb{P}(Y=0|B_i = b_i^0) = \mathbb{P}(Y=1|B_i = b_i^1) = p_i, \forall i \in ${$1, 2$}. This condition is commonly used in the construction of synthetic benchmarks, e.g. colored/coloured-MNIST [1, 2]. The reason why we assume this condition is that it makes the majority/minority split satisfies the label balance criterion, thus we can provide a case where the majority/minority split satisfies the label balance but breaks the falsity exposure, and fail to resolve spurious correlations. If it is not satisfied, the majority/minority split can violate both. We thank the reviewer for raising that point. There was a misuse of the word "Denote"  -> "Suppose", thus making the meaning of this statement vague. We have revised this problem.
>
> **- Another common issue is the usage of conditional probability/distributions given group g, ... in which g is an element.**
>
> In the paper, $g$ equals the event {$ x \in \mathcal{X}| G(x) = i_g $}. As defined in Line 143-145 (153-154 in the new revision), $g$ denotes an element in $\mathcal{G}$, which is the set of inferred groups in the data space. $G$ is defined as the function maps a sample to its group identity, i.e. $G: \mathcal{X} \rightarrow I$ , $G(x) = i_g$ when $x \in g$, where $i_g$ is an identifier of $g$. We have added more clarification for this notation in the new revision.
>
> **- "the main results of this paper seem to center around the construction of the random mapping G ..., but the current presentation of results does not completely bridge the gap between the criterion and the final conclusion and nail the requirement of G in terms of the spurious features.**
>
> We understand that 'main results' here refers to the theoretical result in Theorem 5.1. This theorem is not the central result of this paper, but an additional theoretical result that characteristics the proposed method. We have added more clarification on that point in the revision. As we addressed in the paper, the main results of this paper consist of the following three folds. First, the two criteria on the groups for group-IL to work, which are derived under assumptions on the data generating process introduced in Section 3.  Second, a new algorithm designed under the guidance of the two criteria. It shows the existence of a solution that can provably satisfy the two criteria, and empirically improve the performance of group-IL. Third, experiment results on both synthetic and real-data benchmarks which empirically verify the advantage of SCILL compared with baseline methods and the significance of the two criteria. In the new revision, we further add more ablation study results that empirically verify the criteria.

---

> > ### Author Response · Authors · 2022-08-02
> > **Response to Reviewer i5VX**
> >
> > **- The paper would also benefit from an ablation study to check the importance of the two proposed criteria as well as varying arguments that affect the results including the grouping G and the weights in the objective w.**
> >
> > We thank the reviewer for raising this constructive comment. In Theorem 4.2 and 4.4 we show theoretical results on the importance of the two criteria, which states that they are necessary conditions for group-IL to survive spurious correlations. To verify that, we compare the performance between cases when the criteria are satisfied or not. For example, to show the significance of the falsity exposure criterion, we compare the performance of methods under the case when the label balance criterion is satisfied. On MNLI, we attach the instance reweight step in SCILL to EIIL, obtaining EIIL-lb which satisfies the label balance criterion. As shown in Table 1, EIIL-lb fails to achieve improved performance, which verifies the necessity of falsity exposure.
> >
> > We conduct additional ablation studies on PC-MNIST in the new revision. We experiment with the group splits of SCILL with ground-truth spurious predictor, obtaining SCILL-gt. The better performance of SCILL compared with other baselines verifies the benefit of falsity exposure. For the label balance criterion, we disturb the label balancing weights in SCILL-gt and the significant performance degradation under disturbance shows the importance of label balance. Results are listed in the table below and in the answer to the next comment.
> > |  | ID-test    | Oracle-test | Test-test  |
> > | --------- | ----- | ------ | ----- |
> > | SCILL-IRM | 62.49 | 67.46 | 67.46 |
> > | maj./min. | 50.67 | 63.85 | 50.67 |
> > | SCILL-gt-IRM | 61.12 | 70.19 | 70.91 |
> >
> > **- How strong is the condition in Criterion 4.3 and how would the results be affected if the criterion is violated?**
> >
> > Criterion 4.3 states that if the proportion of samples with label $y, y'$ are non-zero for some group $g, g'$, the equation (2) should hold. The condition does not exist only when the labels of samples in each pair of groups are all different, which rarely happens.
> > Theorem 4.4 theoretically shows the importance of this criterion. Empirically, results of the ablation study on SCILL-gt-IRM below can show how the results will be affected if the criterion is violated.
> >
> > | $p_{err}$ | ID-test    | Oracle-test | Test-test  |
> > | --------- | ----- | ------ | ----- |
> > | 1          | 61.12 | 70.19 | 70.91 |
> > | 1.2       | 61.73 | 64.92  | 63.41 |
> > | 1.5       | 59.57 | 61.27  | 59.26 |
> > | 2          | 58.78 | 59.44 | 56.20 |
> >
> > Here $p_{err}$ is the constant that controls the extent of the unbalance. Specifically, we divide the estimated label proportion of class 0 by $p_{err}$. The results show that violation of the label balance causes significantly worse performance.
> >
> > We hope our response could resolve some concerns of the reviewer and welcome any further questions.
> >
> > References:
> > [1] Arjovsky, Martin, et al. "Invariant risk minimization." arXiv preprint. (2019).
> > [2] Ahmed, Faruk, et al. "Systematic generalisation with group invariant predictions." International Conference on Learning Representations. 2020.

---

> > > ### Comment · Reviewer_i5VX · 2022-08-08
> > > **thank you for the effort**
> > >
> > > I would like to thank the authors for their effort in responding to my comments. I would be happy to increase the score based on the additional results. Meanwhile, my concern about the theoretical component of this paper persists, and I would encourage the authors to consider thoroughly checking the technical results and make sure there is no error as appeared in previous versions of this paper preventing readers from fully examining the results.

---

> > > > ### Author Response · Authors · 2022-08-09
> > > > **Response to Reviewer i5VX**
> > > >
> > > > We thank the reviewer for the reassessment and further suggestions for this paper. As advised by the reviewer,  in the new revision, we have made efforts in the following aspects:
> > > >
> > > > - We have performed several rounds of thorough checks on the main paper and the appendix, correcting all notation issues and typos.
> > > > - We have ensured that every notation in the paper is readable and also clearly defined to avoid ambiguity. For example, in Theorem 4.4 we describe $\mathcal{G}$ as inferred from $(X_{sp}, Y)$ and then give the formal form $\mathcal{G} \subset \sigma( X_{sp} , Y)$. In Section 3, we introduce the definitions of $X_{inv}$, $X_{sp}$ starting from an intuitive understanding of how they can be identified from $X$, and then formally clarify the assumption that there exists a bijection between $X$ and $(X_{inv}, X_{sp})$ after introducing the causal structure. We have refined the definition of some symbols, e.g. $\mathcal{D}$ in Line 124, $\mathbb{P}^e$ and $E$ in Line 129, $f$ in Line 136, $G$ in Line 155, $h$ in Line 182, $f_r^*$ in Line 279.
> > > >
> > > > Furthermore, we have added more clarifications according to several questions raised by the reviewers and have ensured that each statement in this article is fully supported. For example, we revised Section 6.2.2 to add more explanations to the ablation study; we revised the introduction of Theorem 5.1; we added the proof for the statements on causal graphs in Section 3. All the proofs for the statements, theorems, and propositions in this paper can be found in the Appendix. The supplementary materials have included the code to reproduce all the results in the paper and in the responses.
> > > >
> > > > We hope that these revisions address the remaining concern of the reviewer. We thank the reviewer again for taking the time and effort to review this paper.

---

> ### Author Response · Authors · 2022-08-07
> **follow-up**
>
> Dear Reviewer,
>
> We hope that our response has addressed the concerns raised in your review, and that you are willing to reassess your score. If there remain concerns or if you have more questions, we will be more than happy to provide additional clarification. Thank you so much for your time.

---

### Official Review · Reviewer_W2rH · 2022-07-11

**Rating:** 6
**Confidence:** 3
**Ethics Flag:** Yes
**Soundness:** 2 fair
**Presentation:** 3 good
**Contribution:** 2 fair

**Summary:**

They claimed that the existing group invariant learning methods are insufficient to prevent classifiers from depending on spurious correlations in the training set.
In detail, they proposed two criteria on judging such sufficient characteristics.
And then, they showed the limitations of the existing works in terms of the criteria. From this point, they designed a new group invariant learning method, i.e. constructing group and reweighting to meet the criteria. The experimental results support the superiority of the proposed method.

**Questions:**

1. I think that CelebA or CivilComments datasets are suitable for real-world cases. Is there any reasons that this paper does not conduct on these dataset?
2. Group DRO or its variants should be compared. If not necessary, please provide the reasons.
3. There is a hyper-parameter, \lambda, in Equation (3). Is the proposed model sensitive to this value? It would be interesting if there are analysis of the interaction between the value of \lambda and two criteria the authors define.

**Ethics Review Area:**

["I don’t know"]

**Limitations:**

They addressed the limitation and its further direction adequately.

**Strengths And Weaknesses:**

**Strengths**
-
* Two criteria that they designed are suitable for describing the sufficiency of the group invariant learning methods.
* The manuscripts are well-written and easy to follow.
* The analysis for the limitations of the existing works are reasonable and the proposed model is properly designed with this point.

**Weaknesses**
-
* Compared to the analysis until section 4, the proposed model in section 5 is quite simple in terms of the technical novelty.
* In terms of dataset, their experiments are quite limited.
* In terms of baselines, it would be good to compare with Group DRO or its variants.
* The detailed experimental analysis are limited since they just provided the quantitative results.

---

> ### Author Response · Authors · 2022-08-02
> **Response for Reviewer W2rH**
>
> We thank the reviewer for the constructive comments. We respond below to each question.
>
> **- I think that CelebA or CivilComments datasets are suitable for real-world cases. Is there any reasons that this paper does not conduct on these dataset?**
>
> In our experiments we use PC-MNIST as a synthetic dataset for the image classification task, and MNLI-HANS as a real-world case of the NLI task. We agree with the reviewer that CelebA could potentially serve as a real-world image classification benchmark. However, non-trivial effort should be made on the construction of evaluation sets.
>
> The original CelebA dataset contains annotation of 41 attributes of each image. However the split of CelebA in existing works only considers an artificial setting where a single attribute is treated as the spurious feature [1]. Data is divided into four groups according to two binary attributes. Models are evaluated by their minimum accuracy among the four groups. Under this setting, both EI and SCILL satisfy the falsity exposure criterion. The only difference is then the label balancing step in SCILL. However, as already revealed in [2], label balancing improves the worst group performance because CelebA exhibits a large class imbalance. Therefore, empirical findings in this data may not well support the advantage of SCILL.
>
> To compare group-IL methods on that setting, we assume the optimal group splits under EI and SCILL with the ground-truth attributes. The following table shows the result when we conduct model selection by worst group accuracy on the validation set. Consistent with the previous discussion, we can find that SCILL outperforms the EIIL baseline by a large margin on the worst group accuracy.
>
> |  | Average acc | worst group acc |
> | -| --| --|
> | ERM  | 89.82  | 43.89  |
> | EIIL-IRM  | 91.86  | 50.56  |
> | EIIL-REx   | 92.94  | 62.78  |
> | SCILL-IRM | 90.88 | 84.44  |
> | SCILL-REx   | 91.30  | 86.11 |
>
> ResNet18 is used as the model. Optimizer = Adam, learning rate = 1e-4, batch size = 128, l2 regularization parameter = 1e-4, weight decay = 0, epoch=50, annealing epoch=5, invariance penalty weight = 10. Models are evaluated every 2 epochs. Dataset is split following [1].
>
> **- Group DRO or its variants should be compared. If not necessary, please provide the reasons.**
>
> In this paper we focus on invariant learning methods, which is a specific kind of methods focus on the invariance of the feature-conditioned label distribution [3], as we termed in Line 26-27. We also specify their general form in Line 135, equation (1). GroupDRO and its variants are worst-group optimization methods [2], which are another notable kind of algorithms for generalizing to distribution shifts. Our empirical study focus on comparing our new method with existing group-IL methods. Therefore, it is beyond the scope of this paper to compare it with other kinds of methods. Although, it may be an interesting future work to discuss the group criterion for worst-group optimization methods.
>
> **- There is a hyper-parameter, $\lambda$, in Equation (3). Is the proposed model sensitive to this value? It would be interesting if there are analysis of the interaction between the value of  $\lambda$ and two criteria the authors define.**
>
> We thank the reviewer for this constructive comment. This hyperparameter is important for all invariant learning methods, which controls the importance of the invariance penalty in the learning target. Similar to existing works [4], we set a search range for $\lambda$ in model selection. To show the sensitivity, here we list the different test accuracy of model learned with different $\lambda$. With learning rate=5e-3, annealing epoch=100, and penalty=vREx, model selection=Oracle, we have
>
> |        | 0.1   | 1     | 10    | 100   |
> | ----- | ----- | ----- | ----- | ----- |
> | EIIL    |  59.24 | 62.15 | 63.93 | 56.13 |
> | SCILL |   65.95 | 65.89 | 66.36 | 67.15 |
>
> We can find that in this case, the performance of EI and SCILL have different trends with the growth of $\lambda$. We agree with the reviewer that the interaction between the value of $\lambda$ and the two criteria worth further investigation, which will be the future work.
>
> **- The detailed experimental analysis are limited since they just provided the quantitative results.**
>
> We have added more explanations in the new revision on the experiment results and additional ablation studies for verifying the criteria. Specifically, we have made more explanations on the design and comparison of results in the ablation study in Section 6.2.

---

> > ### Author Response · Authors · 2022-08-02
> > **Response for Reviewer W2rH**
> >
> > **- ...the proposed model in section 5 is quite simple in terms of the technical novelty.**
> >
> > In the proposed method, we design a statistical splitting algorithm inspired by the propensity score estimation to satisfy the falsity exposure, and reweight the sample to satisfy label balancing. Although simple, its design follows the guidelines of the two criteria well and empirically achieves significant improvements over the baseline method under various invariant learning penalties. This shows the potential and significance of the proposed criteria. By proposing the algorithm, we show the existence of a solution that can provably satisfy the two criteria, which guarantees the possibility of developing more advanced algorithms in the future.
> >
> > We hope our response could resolve some concerns of the reviewer and welcome any further questions.
> >
> > References:
> >
> > [1] Sagawa, Shiori, et al. "Distributionally robust neural networks for group shifts: On the importance of regularization for worst-case generalization." ICLR. 2019.
> > [2] Idrissi, Badr Youbi, et al. "Simple data balancing achieves competitive worst-group-accuracy." CCLR. PMLR, 2022.
> > [3] Rosenfeld et al."The Risks of Invariant Risk Minimization." ICLR. 2020.
> > [4] Gulrajani et al.. "In Search of Lost Domain Generalization." ICLR. 2020.

---

> > > ### Comment · Reviewer_W2rH · 2022-08-08
> > > **Thanks to the author**
> > >
> > > I have read all comments from the author, and they address my concerns and questions well.
> > >
> > > In this time, one question rises to me.
> > >
> > > In terms of the group inference step via the reference model, I think [1] is also a candidate as a baseline to compare with the proposed model. However, I could not find any comparisons with [1] in the paper.
> > >
> > > Please explain that this is not suitable baseline, if I am wrong.
> > >
> > > Sorry for being late to response. I will positively willing to increase my score if you address this last question.
> > >
> > >
> > > [1] Liu, Evan Z., et al. "Just train twice: Improving group robustness without training group information." International Conference on Machine Learning. PMLR, 2021.

---

> > > > ### Author Response · Authors · 2022-08-08
> > > > **Response for Reviewer W2rH**
> > > >
> > > > We are happy to know that our response has addressed your concerns. We respond below to the new question.
> > > >
> > > > JJT [1] is an algorithm designed to improve the worst-group error in a two-stage manner. They first identify training examples that are misclassified by a standard ERM model, and then train the final model, which is of the same architecture and hyperparameters as the ERM model, by upweighting the examples identified in the first stage. We can see that the design of the group inference step in JJT is related to the upweighting step, but not for the invariant learning methods. This distinction has also been discussed in [2], which introduces that the JJT algorithm "can be seen as a form of distributionally robust optimization where the worst-case distribution only updates once". Therefore, the same as in [2], it is not considered as a baseline method for group-IL in this paper.
> > > >
> > > > Thank you again for taking the time and effort to review this paper.
> > > >
> > > > References:
> > > > [1] Liu, Evan Z., et al. "Just train twice: Improving group robustness without training group information." International Conference on Machine Learning. PMLR, 2021.
> > > > [2] Creager, Elliot, Jörn-Henrik Jacobsen, and Richard Zemel. "Environment inference for invariant learning." International Conference on Machine Learning. PMLR, 2021.

---

> > > > > ### Comment · Reviewer_W2rH · 2022-08-08
> > > > > **Thank you for your efforts**
> > > > >
> > > > > The author address my last questions well. I would like to increase my score as from 5 to 6.
> > > > > With the other reviewers' comments, I highly recommend that the authors clarify the notations/definitions in the paper clearly.

---

> > > > > > ### Author Response · Authors · 2022-08-09
> > > > > > **Response to Reviewer W2rH**
> > > > > >
> > > > > > We thank the reviewer for the reassessment and further suggestions for this paper. As recommended, in the new revision, we have made efforts in the following aspects:
> > > > > >
> > > > > > - We have performed several rounds of thorough checks on the main paper and the appendix, correcting all notation issues and typos.
> > > > > > - We have ensured that every notation in the paper is readable and also clearly defined to avoid ambiguity. For example, in Theorem 4.4 we describe $\mathcal{G}$ as inferred from $(X_{sp}, Y)$ and then give the formal form $\mathcal{G} \subset \sigma( X_{sp} , Y)$. In Section 3, we introduce the definitions of $X_{inv}$, $X_{sp}$ starting from an intuitive understanding of how they can be identified from $X$, and then formally clarify the assumption that there exists a bijection between $X$ and $(X_{inv}, X_{sp})$ after introducing the causal structure. We have refined the definition of some symbols, e.g. $\mathcal{D}$ in Line 124, $\mathbb{P}^e$ and $E$ in Line 129, $f$ in Line 136, $G$ in Line 155, $h$ in Line 182, $f_r^*$ in Line 279.
> > > > > >
> > > > > > Furthermore, we have added more clarifications according to several questions raised by the reviewers and have ensured that each statement in this article is fully supported. For example, we revised Section 6.2.2 to add more explanations to the ablation study; we revised the introduction of Theorem 5.1; we added the proof for the statements on causal graphs in Section 3. All the proofs for the statements, theorems and propositions in this paper can be found in the Appendix. The supplementary materials have included the code to reproduce all the results in the paper and in the responses.

---

> ### Author Response · Authors · 2022-08-07
> **follow-up**
>
> Dear Reviewer,
>
> We hope that our response has addressed the concerns raised in your review, and that you are willing to reassess your score. If there remain concerns or if you have more questions, we will be more than happy to provide additional clarification. Thank you so much for your time.

---

### Official Review · Reviewer_NjkM · 2022-07-11

**Rating:** 7
**Confidence:** 3
**Soundness:** 3 good
**Presentation:** 4 excellent
**Contribution:** 3 good

**Summary:**

The paper proposed the group invariant learning (group-IL) which splits the training data into groups based on spurious correlations. First, they first theoretically analyzed and clarified necessary conditions of group-IL: falsity exposure criterion and label balance criterion. They also discuss that many benchmark datasets violate these conditions. Therefore, they propose a novel methods guided by these conditions. They conduct experiments on image and natural language, and show the effectiveness of their model.

**Questions:**

- It would be nice to modify the caption of each figure of Figure 1, instead of reference numbers.
- Is $h$ in Section 4.1 (e.g. Criterion 4.1) different with $h$ in Line 131, which indicates a classifier?
- How about group inference time compared with baseline?
- Reference model like ERM might be give ambiguous information of spurious correlations. How does it affected?
- Typo
  - Line 72: Classification -> classification
  - Line 207: )) -> )

**Limitations:**

They addressed the limitations in Discussions Section.

**Strengths And Weaknesses:**

### Strengths

- The motivation is reasonable and problem formulation is well constructed with related works.
- Their criterion analysis is well explained, and theoretically supported.
- Proposed model can be applied to most of invariant learning methods.
- The experiment results show the effectiveness of proposed method.
- The paper is well-written.

### Weaknesses

- Their methods are based reference model and they use ERM models. It might be give ambiguous information of spurious correlations.

---

> ### Author Response · Authors · 2022-08-02
> **Response for Reviewer NjkM**
>
> We thank the reviewer for the positive remarks and constructive comments. Please see below for the response to each question:
>
> **- It would be nice to modify the caption of each figure of Figure 1, instead of reference numbers.**
>
> We thank the reviewer for this constructive comment. We have added captions in the new revision.
>
> **- Is h in Section 4.1 (e.g. Criterion 4.1) different with h in Line 131, which indicates a classifier?**
>
> Yes, they are different. $h$ in Section 4.1 denotes an arbitrary (measurable) function, while $h$ in Line 131 denotes the classifier. We have changed the notation of the classifier in Line 131 to $c$ and added more clarification to avoid confusion. Thanks for pointing that out.
>
> **- How about group inference time compared with baseline?**
>
> For 50000 samples (the training set of PCMNIST), it takes 11.246 seconds for the baseline algorithm EI under its default setting (10000 optimization steps). In comparison, our statistical splitting algorithm takes 0.384 seconds in total, which is approximately 30x faster than EI.
>
> **- Reference model like ERM might be give ambiguous information of spurious correlations. How does it affected?**
>
> In our experiments, we follow the design of reference models in existing works. We use ERM model on PC-MNIST, following [1]. On MNLI, we use a biased classifier trained on top of hand-crafted syntactic features, following the design of [2]. Indeed, the ambiguity of reference models impacts the performance of methods depending on it, as shown in [1] for EIIL. How to design better reference models is an important research problem, however out of the scope of this paper.
>
> To show the effect of the quality of the reference model on SCILL, we compare the results on PC-MNIST between SCILL with ERM reference and SCILL with the ground-truth spurious predictor. The following table shows the test accuracy of the two configurations.
>
> | method       | ID    | Oracle | TEV  |
> | ------------ | ----- | ------ | ---- |
> | SCILL-IRM    | 62.49 | 67.46  | 67.46  |
> | SCILL-gt-IRM | 61.12 | 70.19  | 70.91  |
>
> We can see that clear information of spurious correlations can improve the test performance of SCILL. Quality assessment of reference models for group-IL would be an important future study.
>
> **- Typo Line 72: Classification -> classificationLine 207: )) -> )**
>
> Thanks for the correction. We have fixed such typos in the new revision.
>
> References:
>
> [1] Creager, Elliot, Jörn-Henrik Jacobsen, and Richard Zemel. "Environment inference for invariant learning." International Conference on Machine Learning. PMLR, 2021.
> [2] Utama, Prasetya Ajie, Nafise Sadat Moosavi, and Iryna Gurevych. "Mind the Trade-off: Debiasing NLU Models without Degrading the In-distribution Performance." ACL. 2020.

---

> > ### Comment · Reviewer_NjkM · 2022-08-08
> > **Thanks for your response**
> >
> > I would like to thank the author's responses and my concerns have been addressed. I will keep my score as Accept (7).

---

> > > ### Author Response · Authors · 2022-08-09
> > > **Response for Reviewer NjkM**
> > >
> > > We are happy to know that our response has addressed the reviewer's concerns. We would like to thank the reviewer again for taking the time and effort to review this paper.

---

### Meta-Review · Area_Chair_bwG1 · 2022-08-29

**Recommendation:** Accept
**Confidence:** Less certain

**Metareview:**

This paper presents interesting new theory and algorithms to address group-invariant learning in the presence of spurious correlations between unimportant features and the target. A robust discussion between reviewers and authors lends confidence that the paper should be accepted.

I'd encourage the authors to go over language carefully before the camera-ready deadline; there are many small grammatical issues remaining (eg, plurals).

**Award:**

No

---

### Decision · Program_Chairs · 2022-09-14

Accept